# Diversifying Deep Ensembles: A Saliency Map Approach for Enhanced OOD Detection, Calibration, and Accuracy

## Abstract

Deep ensembles are capable of achieving state-of-the-art results in classification and out-of-distribution (OOD) detection. However, their effectiveness is limited due to the homogeneity of learned patterns within ensembles. To overcome this issue, our study introduces Saliency Diversified Deep Ensemble (SDDE), a novel approach that promotes diversity among ensemble members by leveraging saliency maps. Through incorporating saliency map diversification, our method outperforms conventional ensemble techniques and improves calibration in multiple classification and OOD detection tasks. In particular, the proposed method achieves state-of-the-art OOD detection quality, calibration, and accuracy on multiple benchmarks, including CIFAR10/100 and large-scale ImageNet datasets.

## 1 Introduction

In recent years, deep neural networks achieved state-of-the-art results in many computer vision tasks, including object detection Liu et al. (2020), classification Beyer et al. (2020), image retrieval Musgrave et al. (2020), and face recognition Deng et al. (2019). In image classification in particular, DNNs have demonstrated results more accurate than what humans are capable of on several popular benchmarks, such as ImageNet He et al. (2015). However, while these benchmarks often source both training and testing data from a similar distribution, real-world scenarios frequently feature test sets curated independently and under varying conditions Malinin et al. (2021; 2022). This disparity, known as domain shift, can greatly damage the performance of DNNs Gulrajani & Lopez-Paz (2020); Koh et al. (2020); Yao et al. (2022). As such, ensuring robust confidence estimation and out-of-distribution (OOD) detection is paramount for achieving risk-controlled recognition Zhang et al. (2023).

There has been substantial research focusing on confidence estimation and OOD detection in deep learning. Some works consider calibration refinements within softmax classifications Guo et al. (2017), while other authors examine the nuances of Bayesian training Goan & Fookes (2020). In these studies, ensemble methods that use DNNs stand out due to achieving superior outcomes in both confidence estimation and OOD detection Lakshminarayanan et al. (2017); Zhang et al. (2023). The results of these methods can be further improved by diversifying model predictions and adopting novel training paradigms Shui et al. (2018); Pang et al. (2019); Ramé & Cord (2021). However, the works on this subject primarily focus on diversifying the model output without diversifying the feature space.

In this work, we introduce Saliency Diversified Deep Ensemble (SDDE), a novel ensemble training method. This method encourages models to leverage distinct input features for making predictions, as shown in Figure 1. This is implemented through computing saliency maps, or regions of interest, during the training process, and applying a special loss function for diversification. By incorporating these enhancements, we achieve new state-of-the-art (SOTA) results on multiple OpenOOD Zhang et al. (2023) benchmarks in terms of test set accuracy, confidence estimation, and OOD detection quality. Moreover, following past works Hendrycks et al. (2018), we extend our approach by adding OOD data to model training, which made it possible to obtain new SOTA results among methods that utilize OOD data during training.

Class Activation Maps for the models within ensemble

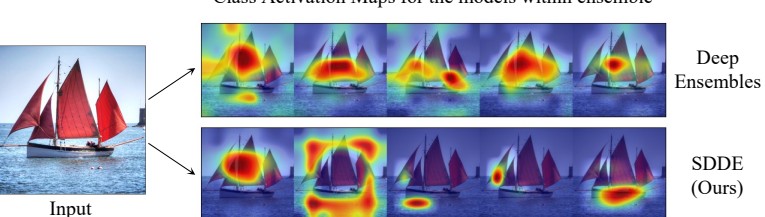

Figure 1: Saliency diversification: compared to Deep Ensembles, the models within the proposed SDDE ensemble use different features for prediction, leading to improved generalization and confidence estimation.

The main contributions of our work are the following:

1. We propose Saliency-Diversified Deep Ensembles (SDDE), a diversification technique that uses saliency maps in order to increase diversity among ensemble models.

2. We achieve new SOTA results in OOD detection, calibration, and classification on the OpenOOD benchmark. In particular, we improved on OOD detection, accuracy, and calibration results on CIFAR10/100 datasets. On ImageNet-1K, we enhance the accuracy and OOD detection scores.

3. We build upon our approach by adding OOD samples during ensemble training. The proposed approach improves ensemble performance and establishes a new SOTA on the CIFAR10 Near/Far and CIFAR100 Near OpenOOD benchmarks.

## 2 PRELIMINARIES

In this section, we will give a brief description of the model and saliency maps estimation approaches used in the proposed SDDE method. Suppose $x \in \mathbb{R}^{CHW}$ is an input image with height $H$, width $W$, and $C$ channels. Let us consider a classification model $f(x) = h(g(x))$, which consists of Convolutional Neural Network (CNN) $g : \mathbb{R}^{CHW} \to \mathbb{R}^{C'H'W'}$ and classifier $h : \mathbb{R}^{C'H'W'} \to \mathbb{R}^L$ with $L$ equal to the number of classes. CNN produces $C'$ feature maps $M^c, c = \overline{1, C'}$ with spatial dimensions $H'$ and $W'$. The output of the classifier is a vector of logits that can be mapped to class probabilities by a softmax layer.

The most common way to evaluate the importance of image pixels for output prediction is by computing saliency maps. In order to build a saliency map of the model, the simplest way is to compute output gradients w.r.t. input features Simonyan et al. (2013):

$$S_{Inp}(x) = \sum_{i=1}^{L} \nabla_x f_i(x). \tag{1}$$

However, previous works show that input gradients produce noisy outputs Simonyan et al. (2013). A more robust approach is to use Class Activation Maps (CAM) Zhou et al. (2016), or their generalization called GradCAM Selvaraju et al. (2017). Both methods utilize spatial directions of the structure of CNNs with bottleneck features. Unlike CAM, GradCAM can be applied at any CNN layer, which is useful for small input images. In GradCAM, the region of interest is estimated by analyzing activations of the feature maps. The weight of each feature map channel is computed as

$$\alpha_c = \frac{1}{H'W'} \sum_{i=1}^{H'} \sum_{j=1}^{W'} \frac{\delta f_y(x)}{\delta M_{i,j}^c}, \tag{2}$$

where $y \in \overline{1, L}$ is an image label. Having the weights of the feature maps, the saliency map is computed as

$$S_{GradCAM}(x, y) = ReLU \left( \sum_{c=1}^{C'} \alpha_c M^c \right),$$ (3)

where ReLU performs element-wise maximum between input and zero. The size of CAM is equal to the size of the feature maps. When visualized, CAMs are resized to match the size of the input image.

## 3 SALIENCY MAPS DIVERSITY AND ENSEMBLE AGREEMENT

Most previous works train ensemble models independently or by diversifying model outputs. Both of these approaches do not take into account the intrinsic algorithms implemented by the models. One way to distinguish classification algorithms is to consider the input features they use. Using saliency maps, which highlights input regions with the largest impact on the model prediction, is a popular technique for identifying these features. In this work, we suggest that ensemble diversity is related to the diversity of saliency maps. To validate this hypothesis, we analyzed the predictions and cosine similarity between saliency maps of the Deep Ensemble model Lakshminarayanan et al. (2017) trained on the MNIST, CIFAR10, CIFAR100, and ImageNet200 datasets. In particular, we compute agreement as a number of models which are consistent with the ensemble prediction. As shown in Figure 2, larger saliency maps similarity usually leads to a larger agreement between models. For more details, please, refer to Appendix F.

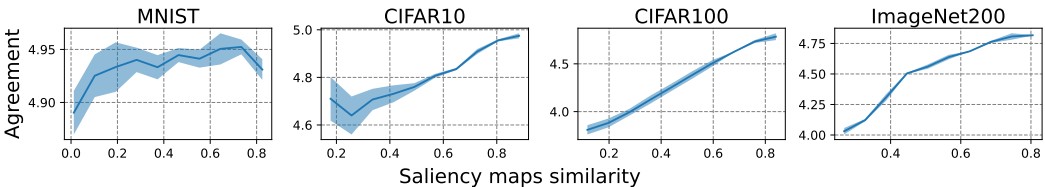

Figure 2: The dependency of ensemble predictions agreement on saliency maps cosine similarity. Saliency maps are computed using GradCAM. Mean and STD values w.r.t. multiple training seeds are reported.

Given this observation, we raise a question whether classification algorithms can be diversified by improving the diversity of saliency maps. We answer this question affirmatively by proposing a new SDDE method for training ensembles. In our experiments, we show the effectiveness of SDDE training in producing diverse high-quality models.

## 4 METHOD

### 4.1 SALIENCY MAPS DIVERSITY LOSS

In Deep Ensemble Lakshminarayanan et al. (2017), the models $f(x; \theta_k), k \in \overline{1, N}$ are trained independently. While there is a source of diversity in weight initialization, this method does not force different models to use different features. Therefore, we explore the following question: how can we train and implement different classification logic for models which rely on different input features? We answer this question by proposing a new loss function, which is motivated by previous research on saliency maps Selvaraju et al. (2017).

The idea behind SDDE is to make the saliency maps Simonyan et al. (2013) of the models as different as possible, as shown in Figure 1. Suppose we have computed a saliency map $S(x, y; \theta_k)$ for each model. The similarity of these models can be measured as a mean cosine similarity between their saliency maps. We thus propose the diversity loss function by computing a mean cosine similarity between the saliency maps of different models:

$$\mathcal{L}_{div}(x, y; \theta_1, \dots, \theta_N) = \frac{2}{N(N-1)} \sum_{k_1 > k_2} \langle S(x, y; \theta_{k_1}), S(x, y; \theta_{k_2}) \rangle .$$ (4)

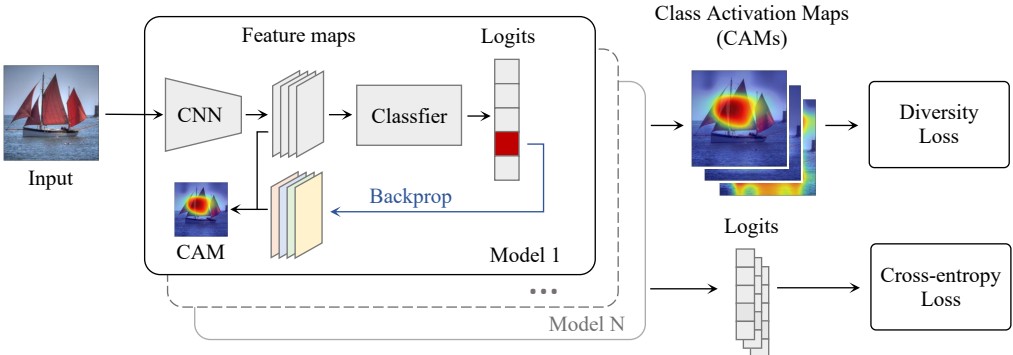

Figure 3: The training pipeline, where we compute saliency maps using the GradCAM method for each model and apply a diversification loss. The final loss also includes the cross-entropy term.

In SDDE, we use GradCAM Selvaraju et al. (2017), as it is more stable and requires less computation than input gradient method Simonyan et al. (2013). As GradCAM is differentiable, the diversity loss can be optimized via gradient descent along with cross-entropy loss.

The diversity loss is combined with cross-entropy during training in a single optimization objective. The final loss has the following form:

$$\mathcal{L}(x, y; \theta_1, \ldots, \theta_N) = \lambda \mathcal{L}_{div}(x, y; \theta_1, \ldots, \theta_N) + \frac{1}{N} \sum_k \mathcal{L}_{CE}(x, y; \theta_k), \tag{5}$$

where $\mathcal{L}_{CE}(x, y; \theta_k)$ is a standard cross-entropy loss and $\lambda$ is a diversity loss weight. Example training algorithm is presented in Appendix H.

## 4.2 AGGREGATION METHODS FOR OOD DETECTION

The original Deep Ensembles approach computes the average over softmax probabilities during inference. The naive approach to OOD detection is to use the Maximum Softmax Probability (MSP) Hendrycks & Gimpel (2016). Suppose there is an ensemble $f(x; \theta_k), k = \overline{1, N}$, where each model $f(x; \theta_k)$ maps input data to the vector of logits. Let us denote softmax outputs for each model as $p_i^k$ and average probabilities as $p_i$:

$$p_i^k(x) = \frac{e^{f_i(x; \theta_k)}}{\sum_j e^{f_j(x; \theta_k)}} = \text{Softmax}_i(f(x; \theta_k)), \tag{6}$$

$$p_i(x) = \frac{1}{N} \sum_k p_i^k(x). \tag{7}$$

The rule of thumb in ensembles is to predict the OOD score based on the maximum output probability:

$$U_{MSP}(x) = \max_i p_i(x). \tag{8}$$

Some recent works follow this approach Abe et al. (2022). However, other works propose to use logits instead of probabilities for OOD detection Hendrycks et al. (2019). In this work, we briefly examine which aggregation is better for ensembles by proposing the Maximum Average Logit (MAL) score. The MAL score extends maximum logit approach for ensembles and is computed as:

$$U_{MAL}(x) = \max_i \frac{1}{N} \sum_k f_i(x; \theta_k). \tag{9}$$

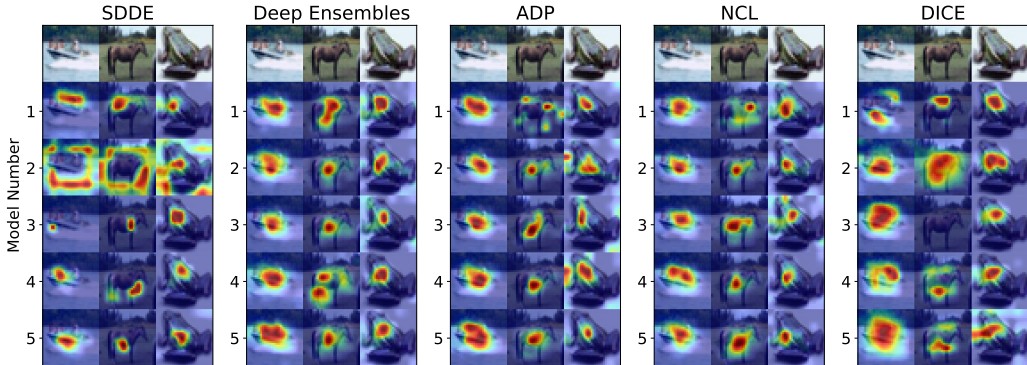

Figure 4: Class Activation Maps (CAMs) for SDDE and the baseline methods. SDDE increases CAMs diversity by focusing on different regions of the images.

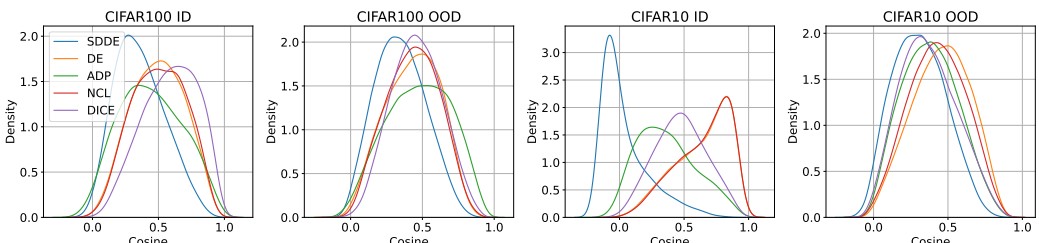

Figure 5: Pairwise distributions of cosine similarities between Class Activation Maps (CAMs) of ensemble models.

## 5 EXPERIMENTS

### 5.1 EXPERIMENTAL SETUP

In our experiments, we follow the experimental setup and training procedure from the OpenOOD benchmark (Zhang et al., 2023). We use ResNet18 (He et al., 2016) for the CIFAR10, CIFAR100 (Krizhevsky & Hinton, 2009), and ImageNet-200 datasets, LeNet (LeCun et al., 1998) for the MNIST dataset (LeCun et al., 1998), and ResNet50 for the ImageNet-1K dataset. All models are trained using the SGD optimizer with a momentum of 0.9. The initial learning rate is set to 0.1 for ResNet18 and LeNet, 0.001 for ResNet50, and then reduced to $10^{-6}$ with the Cosine Annealing scheduler (Loshchilov & Hutter, 2016). Training lasts 50 epochs on MNIST, 100 epochs on CIFAR10 and CIFAR100, 30 epochs on ImageNet-1K, and 90 epochs on ImageNet-200. In contrast to OpenOOD, we perform a multi-seed evaluation with 5 random seeds and report the mean and STD values for each experiment. In ImageNet experiments, we evaluate with 3 seeds. We compare the proposed SDDE method with other ensembling and diversification approaches, namely Deep Ensembles (DE) (Lakshminarayanan et al., 2017), Negative Correlation Learning (NCL) (Shui et al., 2018), Adaptive Diversity Promoting (ADP) (Pang et al., 2019), and DICE diversification loss (Ramé & Cord, 2021). In the default setup, we train ensembles of 5 models. In SDDE, we set the parameter $\lambda$ from the Equation 5 to 0.1 for MNIST, and to 0.005 for CIFAR10/100, ImageNet-200, and ImageNet-1K. The comparison of saliency maps computation approaches can be found in Appendix B. The ablation study on the number of models is presented in Appendix D.

### 5.2 ENSEMBLE DIVERSITY

To demonstrate the effectiveness of the proposed diversification loss, we evaluate the saliency maps for all considered methods, as shown in Figure 4. The ground truth label is not available during inference, so we compute CAMs for the predicted class. It can be seen that SDDE models use different input regions for prediction. To highlight the difference between the methods, we analyze

the pairwise cosine similarities between CAMs of different models in an ensemble. The distributions of cosine similarities are presented in Figure 5. According to the data, the baseline methods reduce similarity compared to Deep Ensemble. However, among all methods, SDDE achieves the lowest cosine similarity. This effect persists even on OOD samples. In the following experiments, we study the benefits of the proposed diversification.

**Prediction-based diversity metrics.** Our saliency-based diversity measure primarily focuses on the variation of saliency maps, which may not fully capture the diversity in the prediction space. However, the benchmarked NCL, ADP, and DICE baselines mainly optimize the diversity in the prediction space. In order to fill this gap, we include additional diversity metrics, such as pairwise disagreement between networks, ratio-error, Q-statistics, correlation coefficient, and mutual information Aksela (2003). The values of metrics for SDDE and baselines are presented in Table 1. It can be seen that SDDE has a lesser impact on the quality of individual models in the ensemble compared to other methods, while achieving moderate diversity values.

Table 1: Diversification metrics. The best result in each column is **bolded**. The DICE method failed to converge on ImageNet.

| Dataset | Method | Diversity | | | | | Error | |
|---|---|---|---|---|---|---|---|---|
| | | Disagreement↑ | Correlation↓ | Q-value↓ | D/S Error Rate↑ | MI↓ | Mean | Ensemble |
| **CIFAR 10** | DE | $4.37_{\pm 0.26}$ | $58.85_{\pm 1.43}$ | $97.38_{\pm 0.31}$ | $0.22_{\pm 0.02}$ | $2.06_{\pm 0.01}$ | $4.89_{\pm 0.13}$ | $3.93_{\pm 0.10}$ |
| | NCL | $4.32_{\pm 0.16}$ | $59.92_{\pm 0.81}$ | $97.53_{\pm 0.17}$ | $0.22_{\pm 0.01}$ | $2.07_{\pm 0.01}$ | $4.95_{\pm 0.13}$ | $4.04_{\pm 0.08}$ |
| | ADP | $\mathbf{4.89}_{\pm 0.06}$ | $\mathbf{57.52}_{\pm 0.39}$ | $\mathbf{97.03}_{\pm 0.09}$ | $\mathbf{0.32}_{\pm 0.01}$ | $\mathbf{2.04}_{\pm 0.00}$ | $5.14_{\pm 0.04}$ | $3.93_{\pm 0.12}$ |
| | DICE | $4.88_{\pm 1.01}$ | $57.61_{\pm 4.15}$ | $96.82_{\pm 1.32}$ | $0.25_{\pm 0.05}$ | $\mathbf{2.04}_{\pm 0.04}$ | $5.24_{\pm 0.57}$ | $4.10_{\pm 0.16}$ |
| | SDDE (Our) | $4.33_{\pm 0.13}$ | $59.29_{\pm 0.95}$ | $97.47_{\pm 0.18}$ | $0.23_{\pm 0.01}$ | $2.07_{\pm 0.01}$ | $\mathbf{4.87}_{\pm 0.11}$ | $\mathbf{3.92}_{\pm 0.11}$ |
| **CIFAR 100** | DE | $20.81_{\pm 2.09}$ | $62.90_{\pm 2.25}$ | $92.70_{\pm 1.42}$ | $0.86_{\pm 0.10}$ | $3.63_{\pm 0.09}$ | $23.12_{\pm 1.28}$ | $19.05_{\pm 0.54}$ |
| | NCL | $20.57_{\pm 0.93}$ | $63.06_{\pm 1.32}$ | $92.86_{\pm 0.70}$ | $0.83_{\pm 0.04}$ | $3.64_{\pm 0.04}$ | $23.00_{\pm 0.47}$ | $19.08_{\pm 0.20}$ |
| | ADP | $\mathbf{25.53}_{\pm 0.28}$ | $63.02_{\pm 0.78}$ | $92.44_{\pm 0.40}$ | $\mathbf{1.78}_{\pm 0.08}$ | $\mathbf{3.43}_{\pm 0.01}$ | $25.13_{\pm 0.12}$ | $18.82_{\pm 0.06}$ |
| | DICE | $21.98_{\pm 0.31}$ | $\mathbf{60.88}_{\pm 0.41}$ | $\mathbf{91.78}_{\pm 0.26}$ | $0.98_{\pm 0.01}$ | $3.59_{\pm 0.01}$ | $23.20_{\pm 0.25}$ | $18.74_{\pm 0.25}$ |
| | SDDE (Our) | $21.00_{\pm 1.34}$ | $61.43_{\pm 1.94}$ | $92.09_{\pm 1.13}$ | $0.86_{\pm 0.07}$ | $3.63_{\pm 0.05}$ | $\mathbf{22.80}_{\pm 0.63}$ | $18.67_{\pm 0.25}$ |
| **ImageNet** | DE | $4.37_{\pm 0.02}$ | $93.65_{\pm 0.03}$ | $99.84_{\pm 0.00}$ | $0.09_{\pm 0.00}$ | $5.99_{\pm 0.00}$ | $25.41_{\pm 0.00}$ | $24.84_{\pm 0.06}$ |
| | NCL | $0.51_{\pm 0.02}$ | $99.31_{\pm 0.03}$ | $100.00_{\pm 0.00}$ | $0.01_{\pm 0.00}$ | $6.17_{\pm 0.00}$ | $25.41_{\pm 0.11}$ | $24.95_{\pm 0.13}$ |
| | ADP | $\mathbf{30.17}_{\pm 0.47}$ | $\mathbf{80.93}_{\pm 0.67}$ | $\mathbf{98.05}_{\pm 0.10}$ | $\mathbf{2.43}_{\pm 0.04}$ | $\mathbf{4.50}_{\pm 0.03}$ | $34.56_{\pm 0.20}$ | $24.90_{\pm 0.01}$ |
| | SDDE (Our) | $4.36_{\pm 0.07}$ | $93.80_{\pm 0.09}$ | $99.85_{\pm 0.00}$ | $0.09_{\pm 0.00}$ | $5.99_{\pm 0.00}$ | $\mathbf{25.39}_{\pm 0.00}$ | $\mathbf{24.80}_{\pm 0.01}$ |

## 5.3 ENSEMBLE ACCURACY AND CALIBRATION

Ensemble methods are most commonly used for improving classification accuracy and prediction calibration metrics. We compare these aspects of SDDE with other ensembling methods by measuring test set classification accuracy, Negative Log-Likelihood (NLL), Expected Calibration Error (ECE) (Naeini et al., 2015), and Brier score (Brier et al., 1950). All metrics are computed after temperature tuning on the validation set (Ashukha et al., 2020). The results are presented in Table 2. It can be seen that the SDDE approach outperforms other methods on CIFAR10 and CIFAR100 in terms of both accuracy and calibration. For the additional experiments with Wide ResNet architecture, which is not part of OpenOOD, please, refer to Appendix C.

Table 2: Accuracy and calibration metrics. The best results for each dataset and metric are **bolded**.

| Dataset | Metric | DE | NCL | ADP | DICE | SDDE (Our) |
|---|---|---|---|---|---|---|
| **MNIST** | NLL ($\times 10$) ↓ | $0.38_{\pm 0.02}$ | $0.36_{\pm 0.02}$ | $\mathbf{0.35}_{\pm 0.02}$ | $0.37_{\pm 0.02}$ | $0.37_{\pm 0.03}$ |
| | ECE ($\times 10^2$) ↓ | $0.21_{\pm 0.07}$ | $\mathbf{0.19}_{\pm 0.07}$ | $0.31_{\pm 0.05}$ | $0.29_{\pm 0.07}$ | $0.28_{\pm 0.08}$ |
| | Brier score ($\times 10^2$) ↓ | $1.95_{\pm 0.09}$ | $1.89_{\pm 0.08}$ | $\mathbf{1.77}_{\pm 0.12}$ | $1.84_{\pm 0.09}$ | $1.92_{\pm 0.15}$ |
| | Accuracy (%) ↑ | $98.72_{\pm 0.07}$ | $98.76_{\pm 0.03}$ | $\mathbf{98.85}_{\pm 0.11}$ | $98.81_{\pm 0.09}$ | $98.73_{\pm 0.14}$ |
| **CIFAR 10** | NLL ($\times 10$) ↓ | $1.30_{\pm 0.02}$ | $1.33_{\pm 0.01}$ | $1.36_{\pm 0.02}$ | $1.45_{\pm 0.03}$ | $\mathbf{1.28}_{\pm 0.01}$ |
| | ECE ($\times 10^2$) ↓ | $0.84_{\pm 0.16}$ | $0.87_{\pm 0.13}$ | $1.00_{\pm 0.14}$ | $1.33_{\pm 0.06}$ | $\mathbf{0.82}_{\pm 0.15}$ |
| | Brier score ($\times 10^2$) ↓ | $5.96_{\pm 0.08}$ | $6.03_{\pm 0.08}$ | $5.99_{\pm 0.07}$ | $6.33_{\pm 0.23}$ | $\mathbf{5.88}_{\pm 0.07}$ |
| | Accuracy (%) ↑ | $96.07_{\pm 0.10}$ | $95.96_{\pm 0.08}$ | $96.07_{\pm 0.12}$ | $95.90_{\pm 0.16}$ | $\mathbf{96.08}_{\pm 0.11}$ |
| **CIFAR 100** | NLL ($\times 10$) ↓ | $7.21_{\pm 0.14}$ | $7.22_{\pm 0.05}$ | $7.54_{\pm 0.03}$ | $7.46_{\pm 0.054}$ | $\mathbf{6.93}_{\pm 0.05}$ |
| | ECE ($\times 10^2$) ↓ | $3.78_{\pm 0.17}$ | $3.86_{\pm 0.23}$ | $3.73_{\pm 0.28}$ | $4.46_{\pm 0.36}$ | $\mathbf{3.48}_{\pm 0.36}$ |
| | Brier score ($\times 10^2$) ↓ | $27.33_{\pm 0.76}$ | $27.34_{\pm 0.33}$ | $27.08_{\pm 0.17}$ | $27.12_{\pm 0.26}$ | $\mathbf{26.64}_{\pm 0.34}$ |
| | Accuracy (%) ↑ | $80.95_{\pm 0.54}$ | $80.92_{\pm 0.20}$ | $81.18_{\pm 0.06}$ | $81.26_{\pm 0.25}$ | $\mathbf{81.33}_{\pm 0.28}$ |

## 5.4 OOD Detection

We evaluate SDDE's OOD detection ability using the OpenOOD benchmark. Since SDDE does not use external data for training, we compare it to ensembles that only use in-distribution data. The results are presented in Table 3. It can be seen that SDDE achieves SOTA results in almost all cases, including the total scores on near and far tests. For the additional analysis on different logit aggregation methods, please, refer to Appendix E.

Table 3: OOD detection results. All methods are trained on the ID dataset and tested on multiple OOD sources. Mean and STD AUROC values are reported. The best results among different methods are **bolded**.

| ID | Method | Near OOD | | Far OOD | | | | Total Near | Total Far |
|---|---|---|---|---|---|---|---|---|---|
| | | Fashion MNIST | NotMNIST | CIFAR10 | TIN | Texture | Places365 | | |
| MNIST | DE | $95.34_{\pm0.53}$ | $89.56_{\pm0.84}$ | $99.06_{\pm0.15}$ | $98.93_{\pm0.18}$ | $96.32_{\pm0.79}$ | $99.06_{\pm0.14}$ | $92.45_{\pm0.60}$ | $98.34_{\pm0.31}$ |
| | NCL | $95.27_{\pm0.42}$ | $89.19_{\pm0.69}$ | $99.06_{\pm0.11}$ | $98.97_{\pm0.12}$ | $97.17_{\pm0.45}$ | $99.03_{\pm0.11}$ | $92.23_{\pm0.53}$ | $98.56_{\pm0.19}$ |
| | ADP | $96.49_{\pm0.67}$ | $90.98_{\pm0.65}$ | $99.48_{\pm0.18}$ | $99.38_{\pm0.21}$ | $97.52_{\pm0.77}$ | $99.44_{\pm0.15}$ | $93.74_{\pm0.48}$ | $98.96_{\pm0.32}$ |
| | DICE | $95.85_{\pm0.22}$ | $89.88_{\pm1.51}$ | $99.22_{\pm0.19}$ | $99.10_{\pm0.22}$ | $97.52_{\pm1.05}$ | $99.20_{\pm0.17}$ | $92.87_{\pm0.67}$ | $98.76_{\pm0.41}$ |
| | **SDDE (Our)** | $\mathbf{98.85}_{\pm0.38}$ | $\mathbf{94.27}_{\pm1.32}$ | $\mathbf{99.88}_{\pm0.03}$ | $\mathbf{99.84}_{\pm0.03}$ | $\mathbf{99.97}_{\pm0.01}$ | $\mathbf{99.82}_{\pm0.04}$ | $\mathbf{96.56}_{\pm0.67}$ | $\mathbf{99.88}_{\pm0.02}$ |
| | | CIFAR100 | TIN | MNIST | SVHN | Texture | Places365 | | |
| CIFAR10 | DE | $90.31_{\pm0.23}$ | $91.84_{\pm0.16}$ | $95.35_{\pm0.53}$ | $95.05_{\pm0.47}$ | $92.52_{\pm0.30}$ | $91.29_{\pm0.59}$ | $91.07_{\pm0.17}$ | $93.55_{\pm0.20}$ |
| | NCL | $90.51_{\pm0.12}$ | $91.93_{\pm0.08}$ | $95.09_{\pm0.29}$ | $94.81_{\pm0.46}$ | $92.18_{\pm0.34}$ | $92.00_{\pm0.26}$ | $91.22_{\pm0.09}$ | $93.52_{\pm0.17}$ |
| | ADP | $89.96_{\pm0.25}$ | $91.62_{\pm0.13}$ | $95.41_{\pm0.24}$ | $94.66_{\pm0.40}$ | $92.55_{\pm0.12}$ | $91.38_{\pm0.53}$ | $90.79_{\pm0.17}$ | $93.50_{\pm0.16}$ |
| | DICE | $89.09_{\pm0.79}$ | $90.89_{\pm0.48}$ | $94.46_{\pm1.10}$ | $94.89_{\pm0.49}$ | $92.36_{\pm0.45}$ | $89.84_{\pm1.14}$ | $89.99_{\pm0.63}$ | $92.89_{\pm0.47}$ |
| | **SDDE (Our)** | $\mathbf{91.20}_{\pm0.13}$ | $\mathbf{92.92}_{\pm0.17}$ | $\mathbf{96.77}_{\pm0.45}$ | $\mathbf{95.74}_{\pm0.36}$ | $\mathbf{92.69}_{\pm0.62}$ | $\mathbf{93.19}_{\pm0.62}$ | $\mathbf{92.06}_{\pm0.13}$ | $\mathbf{94.60}_{\pm0.17}$ |
| | | CIFAR10 | TIN | MNIST | SVHN | Texture | Places365 | | |
| CIFAR100 | DE | $80.83_{\pm0.55}$ | $84.28_{\pm0.52}$ | $80.24_{\pm1.43}$ | $81.09_{\pm1.27}$ | $80.16_{\pm0.41}$ | $80.96_{\pm0.26}$ | $82.55_{\pm0.51}$ | $80.61_{\pm0.51}$ |
| | NCL | $81.11_{\pm0.20}$ | $84.48_{\pm0.19}$ | $79.66_{\pm0.66}$ | $80.91_{\pm1.99}$ | $80.30_{\pm0.52}$ | $81.24_{\pm0.17}$ | $82.79_{\pm0.18}$ | $80.53_{\pm0.66}$ |
| | ADP | $81.12_{\pm0.19}$ | $84.85_{\pm0.22}$ | $79.54_{\pm0.76}$ | $82.84_{\pm2.16}$ | $81.80_{\pm0.47}$ | $81.28_{\pm0.11}$ | $82.98_{\pm0.18}$ | $81.37_{\pm0.54}$ |
| | DICE | $81.42_{\pm0.18}$ | $84.94_{\pm0.21}$ | $\mathbf{83.37}_{\pm1.11}$ | $82.40_{\pm1.96}$ | $81.43_{\pm0.39}$ | $81.41_{\pm0.27}$ | $83.18_{\pm0.14}$ | $82.15_{\pm0.85}$ |
| | **SDDE (Our)** | $\mathbf{81.97}_{\pm0.10}$ | $\mathbf{85.34}_{\pm0.16}$ | $81.86_{\pm1.80}$ | $\mathbf{83.40}_{\pm1.03}$ | $\mathbf{82.67}_{\pm0.35}$ | $\mathbf{81.63}_{\pm0.18}$ | $\mathbf{83.65}_{\pm0.09}$ | $\mathbf{82.39}_{\pm0.56}$ |

## 5.5 ImageNet Results

In addition to the above-mentioned datasets, we conduct experiments on the large-scale ImageNet-1K benchmark from OpenOOD. The results for accuracy, calibration, and OOD detection are presented in Table 4. It can be observed that on ImageNet-1K SDDE achieves the best accuracy and OOD detection score among all the methods. Furthermore, SDDE achieves calibration scores comparable to the best-performing method in each column. The DICE training failed to converge in this setup, and we excluded it from the comparison.

Table 4: ImageNet results. The best-performing method in each column is **bolded**.

| Dataset | Method | Near OOD | Far OOD | NLL ($\times10$) | ECE ($\times10^2$) | Brier ($\times10^2$) | Accuracy (%) |
|---|---|---|---|---|---|---|---|
| **ImageNet** | DE | $75.02_{\pm0.06}$ | $82.72_{\pm0.26}$ | $9.79_{\pm0.00}$ | $1.54_{\pm0.06}$ | $\mathbf{34.40}_{\pm0.01}$ | $75.16_{\pm0.06}$ |
| | NCL | $75.06_{\pm0.05}$ | $82.85_{\pm0.17}$ | $9.83_{\pm0.00}$ | $\mathbf{1.51}_{\pm0.02}$ | $34.51_{\pm0.14}$ | $75.05_{\pm0.13}$ |
| | ADP | $\mathbf{75.26}_{\pm0.01}$ | $82.62_{\pm0.08}$ | $9.95_{\pm0.01}$ | $1.95_{\pm0.11}$ | $34.54_{\pm0.08}$ | $75.10_{\pm0.01}$ |
| | **SDDE (Our)** | $\mathbf{75.26}_{\pm0.02}$ | $\mathbf{85.98}_{\pm0.61}$ | $\mathbf{9.79}_{\pm0.01}$ | $1.55_{\pm0.05}$ | $34.43_{\pm0.05}$ | $\mathbf{75.20}_{\pm0.01}$ |

## 5.6 Distribution Shifts

In addition to OOD detection, accuracy, and calibration, we evaluate SDDE's performance on datasets with distribution shifts (OOD generalization), such as the CIFAR10-C and CIFAR100-C datasets Hendrycks & Dietterich (2019). The accuracy and calibration metrics are reported in Table 5. These results demonstrate that SDDE outperforms the baselines in both accuracy and calibration metrics on CIFAR10-C and achieves accuracy on par with the best-performing method while improving the calibration metrics on CIFAR100-C.

## 5.7 Leveraging OOD Data for Training

In some applications, an unlabeled OOD sample can be provided during training to further improve OOD detection quality. According to previous studies Zhang et al. (2023), Outlier Exposure (OE) Hendrycks et al. (2018) is one of the most accurate methods on CIFAR10/100 and ImageNet-200

Table 5: Accuracy and calibration metrics on corrupted datasets.

| Method | CIFAR10-C | | | | CIFAR100-C | | | |
|---|---|---|---|---|---|---|---|---|
| | NLL ($\times 10$) | ECE ($\times 10^2$) | Brier ($\times 10^2$) | Accuracy | NLL ($\times 10$) | ECE ($\times 10^2$) | Brier ($\times 10^2$) | Accuracy |
| DE | 1.06 $\pm 0.01$ | **4.67** $\pm 0.12$ | 44.68 $\pm 0.58$ | 67.74 $\pm 0.38$ | 12.91 $\pm 0.10$ | 4.74 $\pm 0.26$ | 43.28 $\pm 0.59$ | 68.55 $\pm 0.55$ |
| NCL | 1.06 $\pm 0.01$ | 4.83 $\pm 0.37$ | 44.98 $\pm 0.49$ | 67.43 $\pm 0.40$ | 12.83 $\pm 0.12$ | 4.69 $\pm 0.10$ | 43.07 $\pm 0.36$ | 68.71 $\pm 0.30$ |
| ADP | 1.06 $\pm 0.02$ | 4.67 $\pm 0.12$ | 44.64 $\pm 0.58$ | 67.81 $\pm 0.38$ | 13.14 $\pm 0.11$ | 4.97 $\pm 0.10$ | 42.61 $\pm 0.32$ | **69.20** $\pm 0.22$ |
| DICE | 1.10 $\pm 0.02$ | 7.41 $\pm 0.53$ | 45.86 $\pm 0.70$ | 67.14 $\pm 0.60$ | 13.57 $\pm 0.09$ | 5.23 $\pm 0.26$ | 44.40 $\pm 0.50$ | 67.91 $\pm 0.48$ |
| **SDDE (Ours)** | **1.04** $\pm 0.01$ | 4.84 $\pm 0.38$ | **44.32** $\pm 0.31$ | **67.96** $\pm 0.28$ | **12.58** $\pm 0.08$ | **4.00** $\pm 0.09$ | **42.43** $\pm 0.32$ | 69.08 $\pm 0.27$ |

Table 6: Evaluation results for methods trained with OOD data.

| Dataset | Method | Near OOD | Far OOD | NLL ($\times 10$) | ECE ($\times 10^2$) | Brier ($\times 10^2$) | Accuracy (%) |
|---|---|---|---|---|---|---|---|
| **CIFAR 10** | OE Single | 94.82 ± 0.21 | 96.00 ± 0.13 | 2.85 ± 0.26 | 5.88 ± 0.97 | 11.62 ± 0.98 | 94.63 ± 0.26 |
| | OE Ensemble | **96.23 ± 0.08** | 97.45 ± 0.15 | 2.65 ± 0.05 | **4.86 ± 0.25** | 10.89 ± 0.19 | 95.84 ± 0.19 |
| | SDDE$_{OOD}$ (Ours) | 96.22 ± 0.08 | **97.56 ± 0.07** | **2.64 ± 0.02** | 4.95 ± 0.15 | **10.86 ± 0.15** | **95.87 ± 0.02** |
| **CIFAR 100** | OE Single | 88.30 ± 0.10 | 81.41 ± 1.49 | **9.77 ± 0.28** | 8.41 ± 0.47 | **34.11 ± 0.73** | 76.84 ± 0.42 |
| | OE Ensemble | 89.61 ± 0.04 | **84.53 ± 0.72** | 9.93 ± 0.11 | **8.35 ± 0.20** | 34.34 ± 0.39 | **80.65 ± 0.30** |
| | SDDE$_{OOD}$ (Ours) | **89.70 ± 0.22** | 85.47 ± 1.76 | 10.13 ± 0.28 | 8.70 ± 0.31 | 34.86 ± 0.60 | 80.30 ± 0.25 |
| **ImageNet-200** | OE Single | 81.87 ± 0.16 | 86.77 ± 0.06 | 21.88 ± 0.46 | 37.31 ± 1.10 | 54.67 ± 1.11 | 77.57 ± 0.28 |
| | OE Ensemble | 84.41 ± 0.15 | 89.45 ± 0.26 | 21.71 ± 0.07 | **36.68 ± 0.14** | 54.37 ± 0.12 | **83.36 ± 0.07** |
| | SDDE$_{OOD}$ (Ours) | **84.46 ± 0.07** | **89.57 ± 0.11** | **21.63 ± 0.29** | 36.71 ± 0.79 | **54.17 ± 0.69** | 83.36 ± 0.08 |

datasets. The idea behind OE is to make predictions on OOD data as close to the uniform distribution as possible. This is achieved by minimizing cross-entropy between the uniform distribution and the output of the model on OOD data:

$$\mathcal{L}_{OE}(x; \theta) = -\frac{1}{C} \sum_i \log \mathrm{Softmax}_i(f(x; \theta)). \qquad (10)$$

Given an unlabeled OOD sample $\overline{x}$, we combine OE loss with SDDE, leading to the following objective:

$$\mathcal{L}_{OOD}(x, y, \overline{x}; \theta_1, \ldots, \theta_N) = \mathcal{L}(x, y; \theta_1, \ldots, \theta_N) + \beta \frac{1}{N} \mathcal{L}_{OE}(\overline{x}; \theta_k), \qquad (11)$$

We follow the original OE implementation and set $\beta$ to 0.5. We call the final method SDDE$_{OOD}$.

The OpenOOD implementation of OE includes only a single model. In order to have a fair comparison, we train an ensemble of 5 OE models and average their predictions.

The results are presented in Table 6. It can be seen that SDDE$_{OOD}$ achieves higher OOD detection accuracy compared to the current SOTA OE method in OpenOOD. Following SDDE experiments, we evaluate SDDE$_{OOD}$ calibration and accuracy. SDDE$_{OOD}$ demonstrates competitive accuracy and calibration scores on all benchmark datasets.

## 6 RELATED WORK

**Confidence Estimation.** DNNs are prone to overfitting, which limits their ability to generalize and predict confidence Guo et al. (2017). Multiple works attempted to address this issue. A simple approach to confidence estimation is to take the probability of a predicted class on the output of the softmax layer Hendrycks & Gimpel (2016). Several authors proposed improvements to this method for either better confidence estimation or higher OOD detection accuracy.

Some works studied activation statistics between layers to detect anomalous behavior Sun et al. (2021). Another approach is to use Bayesian training, which usually leads to improved confidence prediction at the cost of accuracy Goan & Fookes (2020). Other works attempted to use insights from classical machine learning, such as KNNs Sun et al. (2022) and ensembles Lakshminarayanan et al. (2017). It was shown that ensembles reduce overfitting and produce confidence estimates, which outperforms most existing methods Zhang et al. (2023). In this work, we further improve deep ensembles by applying a new training algorithm.

**Ensembles Diversification.** According to previous works, ensemble methods produce SOTA results in popular classification Lakshminarayanan et al. (2017); Ramé & Cord (2021) and OOD detection tasks Zhang et al. (2023). It was shown that the quality of the ensemble largely depends on the

diversity of underlying models Stickland & Murray (2020). Some works improve diversity by implementing special loss functions on predicted probabilities. The idea is to make different predicted distributions for different models. The NCL loss Shui et al. (2018) reduces correlations between output probabilities, but can negatively influence the prediction of the correct class. The ADP Pang et al. (2019) solves this problem by diversifying only the probabilities of alternative classes. On the other hand, the DICE Ramé & Cord (2021) loss reduces dependency between bottleneck features of multiple models. While previous works reduce the correlations of either model predictions or bottleneck features, we directly diversify the input features used by the models within an ensemble, which further improves OOD detection and calibration.

**Out-of-Distribution Detection**   Risk-controlled recognition poses the problem of detecting out-of-distribution (OOD) data, i.e. data with distribution different from the training set, or data with unknown classes Zhang et al. (2023). Multiple OOD detection methods were proposed for deep models Sun et al. (2021); Hendrycks et al. (2019). Despite the progress made on single models, Deep Ensembles Lakshminarayanan et al. (2017) use a traditional Maximum Softmax Probability (MSP) Hendrycks & Gimpel (2016) approach to OOD detection. In this work, we introduce a novel ensembling method that offers enhanced OOD detection capabilities. Furthermore, we extend the Maximum Logit Score (MLS) Hendrycks et al. (2019) for ensemble application, underscoring its advantages over MSP.

## 7   LIMITATIONS

While the Saliency-Diversified Deep Ensembles (SDDE) method can offer promising advancements in OOD detection, calibration, and classification, there are some inherent limitations to consider. Firstly, while incorporating OOD samples during ensemble training can be beneficial, this approach could raise questions about the quality, diversity, and source of these samples, as well as whether such results would generalize when faced with unseen OOD categories. Additionally, while achieving SOTA results on the OpenOOD benchmark is notable, benchmarks constantly evolve, and the real-world applicability and robustness of the method across varied scenarios and datasets is yet to be extensively tested. Nevertheless, while it is important to take these limitations into account, they do not invalidate our conclusions. Instead, they qualify our results, serving as potential starting points for future research.

## 8   CONCLUSION

In this work, we proposed SDDE, a novel ensembling method for classification and OOD detection. SDDE forces the models within the ensemble to use different input features for prediction, which increases ensemble diversity. According to our experiments, SDDE performs better than several popular ensembles on the CIFAR10, CIFAR100, and ImageNet-1K datasets. At the same time, SDDE outperforms other methods in OOD detection on the OpenOOD benchmark. Improved confidence estimation and OOD detection make SDDE a valuable tool for risk-controlled recognition. We further generalized SDDE for training with OOD data and achieved SOTA results on the OpenOOD benchmark.

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

## A    ADVERSARIAL LOSS IN DEEP ENSEMBLES

The original Deep Ensemble approach Lakshminarayanan et al. (2017) applied an extra adversarial loss, which is not present in OpenOOD. Therefore, we evaluate the impact of this loss on accuracy, calibration and OOD detection scores. The adversarial loss is inspired by adversarial samples generation Goodfellow et al. (2014) and has the following form:

$$\mathcal{L}_{adv} = \frac{1}{N} \sum_k \mathcal{L}_{CE}(\hat{x}_k, y, \theta_k), \tag{12}$$

where $\hat{x}_k$ is an adversarial version of the input image $x$ with label $y$, computed as

$$\hat{x}_k = x + \epsilon \operatorname{sign}\left(\nabla_x \mathcal{L}_{CE}(x, y, \theta_k)\right). \tag{13}$$

We follow the original implementation and set $\epsilon$ equal to $0.01$ of the input data's dynamic range.

We evaluated the adversarial loss for the CIFAR-10 and CIFAR-100 datasets. According to our results, presented in Table 7, adversarial loss slightly reduces the quality of the ensemble. We conclude that adversarial loss improves ensemble robustness to adversarial samples, but negatively affects accuracy and OOD detection.

Table 7: Diversity loss ablation study.

| Dataset | Method | Near ODD ↑ | Far OOD ↑ | NLL ($\times 10$) ↓ | ECE ($\times 10^2$) ↓ | Brier score ($\times 10^2$) ↓ | Accuracy (%) ↑ |
|---|---|---|---|---|---|---|---|
| MNIST | DE | $92.45 \pm 0.60$ | $98.34 \pm 0.31$ | $0.38 \pm 0.02$ | $\mathbf{0.21} \pm 0.06$ | $1.95 \pm 0.09$ | $98.72 \pm 0.07$ |
| | DE with $\mathcal{L}_{adv}$ | $\mathbf{95.63} \pm 0.46$ | $\mathbf{99.55} \pm 0.02$ | $\mathbf{0.34} \pm 0.01$ | $0.26 \pm 0.04$ | $\mathbf{1.76} \pm 0.05$ | $\mathbf{98.84} \pm 0.06$ |
| CIFAR10 | DE | $\mathbf{91.07} \pm 0.17$ | $\mathbf{93.55} \pm 0.20$ | $\mathbf{1.30} \pm 0.02$ | $\mathbf{0.84} \pm 0.16$ | $\mathbf{5.96} \pm 0.08$ | $\mathbf{96.07} \pm 0.10$ |
| | DE with $\mathcal{L}_{adv}$ | $90.84 \pm 0.13$ | $92.12 \pm 0.22$ | $2.02 \pm 0.04$ | $0.93 \pm 0.09$ | $9.60 \pm 0.15$ | $93.50 \pm 0.21$ |
| CIFAR100 | DE | $\mathbf{82.55} \pm 0.51$ | $\mathbf{80.61} \pm 0.51$ | $\mathbf{7.21} \pm 0.14$ | $3.78 \pm 0.17$ | $\mathbf{27.33} \pm 0.76$ | $\mathbf{80.95} \pm 0.54$ |
| | DE with $\mathcal{L}_{adv}$ | $80.87 \pm 0.60$ | $76.87 \pm 0.99$ | $10.06 \pm 0.05$ | $\mathbf{3.11} \pm 0.79$ | $37.00 \pm 0.35$ | $72.74 \pm 0.39$ |

## B    INPUT GRADIENTS VS GRADCAM

In this work, we diversify saliency maps computed by GradCAM. We also evaluate saliency maps constructed by taking gradients w.r.t. the input image, i.e. *input gradients* Simonyan et al. (2013). The results from Table 8 demonstrate the superiority of GradCAM to input gradients in terms of OOD detection and classification accuracy.

Another difference between input gradients and GradCAM lies in computation speed. Diversity loss requires two backward passes: one for making a saliency map, and another for loss optimization. The complexity of both passes depends on the method used. Computing gradients w.r.t. the input images requires a backward pass through all layers, while GradCAM stops when the feature extraction layer is reached. This leads to a difference in training speed, making GradCAM about 2.4 times faster than input gradients. As GradCAM outperforms input gradients in both accuracy and training speed, we decided to use it as a default choice for ensemble diversification.

Table 8: Comparison of saliency map computation algorithms.

| Dataset | Method | Near ODD ↑ | Far OOD ↑ | NLL ($\times 10$) ↓ | ECE ($\times 10^2$) ↓ | Brier score ($\times 10^2$) ↓ | Accuracy (%) ↑ |
|---|---|---|---|---|---|---|---|
| MNIST | Inp. grad. | $92.04 \pm 0.57$ | $98.59 \pm 0.30$ | $\mathbf{0.37} \pm 0.02$ | $\mathbf{0.28} \pm 0.40$ | $\mathbf{1.92} \pm 0.09$ | $\mathbf{98.79} \pm 0.05$ |
| | GradCAM | $\mathbf{96.56} \pm 0.67$ | $\mathbf{99.88} \pm 0.02$ | $0.37 \pm 0.03$ | $0.28 \pm 0.08$ | $1.92 \pm 0.15$ | $98.73 \pm 0.14$ |
| CIFAR10 | Inp. grad. | $91.18 \pm 0.13$ | $93.56 \pm 0.24$ | $1.35 \pm 0.05$ | $\mathbf{0.76} \pm 0.12$ | $6.23 \pm 0.34$ | $95.89 \pm 0.21$ |
| | GradCAM | $\mathbf{92.06} \pm 0.13$ | $\mathbf{94.60} \pm 0.17$ | $\mathbf{1.28} \pm 0.01$ | $0.82 \pm 0.15$ | $\mathbf{5.88} \pm 0.07$ | $\mathbf{96.08} \pm 0.11$ |
| CIFAR100 | Inp. grad. | $82.80 \pm 0.18$ | $80.06 \pm 0.58$ | $7.14 \pm 0.03$ | $3.76 \pm 0.26$ | $27.06 \pm 0.27$ | $81.06 \pm 0.14$ |
| | GradCAM | $\mathbf{83.69} \pm 0.08$ | $\mathbf{82.39} \pm 0.50$ | $\mathbf{6.93} \pm 0.05$ | $\mathbf{3.45} \pm 0.35$ | $\mathbf{26.66} \pm 0.33$ | $\mathbf{81.27} \pm 0.24$ |

## C   WIDE RESNET-28-10 EXPERIMENTS

In addition to OpenOOD benchmarks, we conducted experiments with the Wide ResNet architecture. The results in Table 9 show that SDDE with WRN-28-10 outperforms other methods in terms of accuracy, calibration, and OOD detection scores.

Table 9: Wide ResNet 28-10 results.

| Dataset | Method | Near OOD | Far OOD | NLL | ECE | Brier | Accuracy |
|---|---|---|---|---|---|---|---|
| CIFAR 10 | DE | $90.31 _{\pm 0.18}$ | $93.70 _{\pm 0.18}$ | $11.87 _{\pm 0.10}$ | $\mathbf{0.94} _{\pm \mathbf{0.14}}$ | $5.27 _{\pm 0.11}$ | $96.59 _{\pm 0.15}$ |
| | NCL | $90.26 _{\pm 0.37}$ | $93.44 _{\pm 0.20}$ | $12.20 _{\pm 0.34}$ | $0.78 _{\pm 0.16}$ | $5.39 _{\pm 0.24}$ | $96.48 _{\pm 0.19}$ |
| | ADP | $88.81 _{\pm 0.49}$ | $92.86 _{\pm 0.41}$ | $12.77 _{\pm 0.29}$ | $0.75 _{\pm 0.12}$ | $5.39 _{\pm 0.15}$ | $96.57 _{\pm 0.12}$ |
| | DICE | $87.37 _{\pm 0.39}$ | $91.52 _{\pm 0.20}$ | $14.44 _{\pm 0.32}$ | $1.09 _{\pm 0.10}$ | $6.03 _{\pm 0.18}$ | $96.17 _{\pm 0.16}$ |
| | SDDE (Our) | $\mathbf{90.40} _{\pm \mathbf{0.23}}$ | $\mathbf{94.30} _{\pm \mathbf{0.22}}$ | $\mathbf{11.80} _{\pm \mathbf{0.17}}$ | $\mathbf{0.94} _{\pm \mathbf{0.15}}$ | $\mathbf{5.16} _{\pm \mathbf{0.11}}$ | $\mathbf{96.65} _{\pm \mathbf{0.06}}$ |
| CIFAR 100 | DE | $83.03 _{\pm 0.20}$ | $81.27 _{\pm 0.69}$ | $66.32 _{\pm 0.47}$ | $3.55 _{\pm 0.30}$ | $24.48 _{\pm 0.35}$ | $82.99 _{\pm 0.33}$ |
| | NCL | $83.02 _{\pm 0.23}$ | $80.47 _{\pm 0.20}$ | $66.46 _{\pm 0.54}$ | $3.49 _{\pm 0.30}$ | $24.45 _{\pm 0.09}$ | $83.05 _{\pm 0.11}$ |
| | ADP | $82.48 _{\pm 0.26}$ | $79.78 _{\pm 0.98}$ | $70.75 _{\pm 0.55}$ | $3.50 _{\pm 0.11}$ | $24.85 _{\pm 0.33}$ | $83.02 _{\pm 0.22}$ |
| | DICE | $82.17 _{\pm 0.78}$ | $80.11 _{\pm 1.17}$ | $72.36 _{\pm 0.97}$ | $3.99 _{\pm 0.12}$ | $25.46 _{\pm 0.57}$ | $82.56 _{\pm 0.56}$ |
| | SDDE (Our) | $\mathbf{83.54} _{\pm \mathbf{0.12}}$ | $\mathbf{82.81} _{\pm \mathbf{0.42}}$ | $\mathbf{65.66} _{\pm \mathbf{0.20}}$ | $\mathbf{3.39} _{\pm \mathbf{0.09}}$ | $\mathbf{24.17} _{\pm \mathbf{0.14}}$ | $\mathbf{83.34} _{\pm \mathbf{0.15}}$ |

## D   NUMBER OF MODELS

The key parameter of any ensemble is its number of models. A large ensemble usually provides a better confidence estimate, but its size negatively affects the computation speed. In order to determine the best number of models to use without incurring losses, we analyze the dependency of quality on ensemble size. The results for classification and OOD detection quality are presented in Figure 6. It can be seen that SDDE effectively uses multiple models, leading to better results for all ensemble sizes, starting from 3 in the CIFAR-10 and CIFAR-100 Near and Far setups.

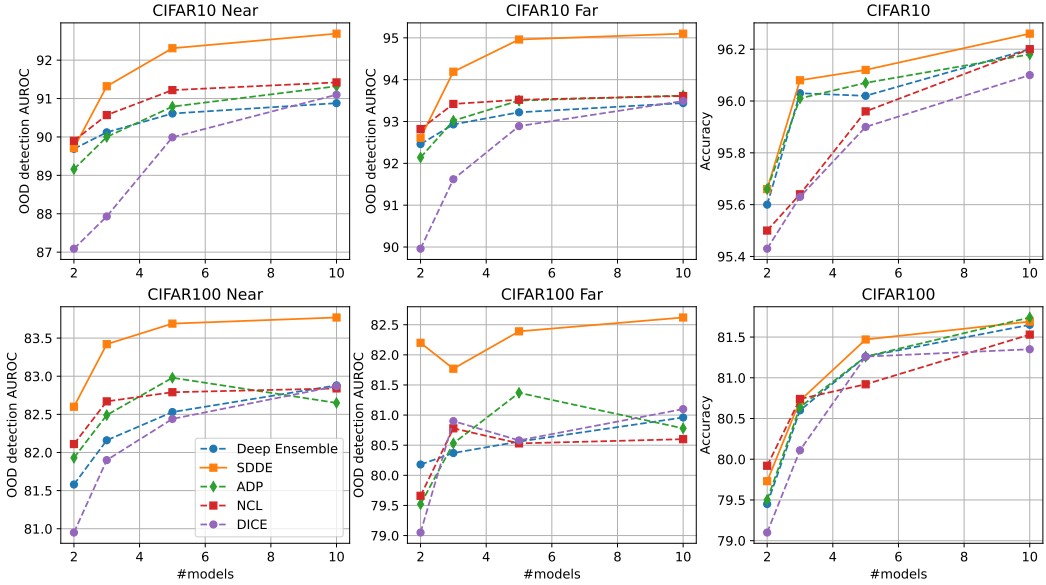

Figure 6: Classification accuracy and OOD detection quality depending on ensemble size.

# E    LOGIT AGGREGATION ABLATION

In Section 4, we introduced logit-based prediction aggregation for the proposed SDDE approach. Although a single-model maximal logit score can be used for OOD detection Hendrycks et al. (2019), it is still unclear whether logit averaging is applicable for ensemble OOD detection. To determine this, we analyze the outputs of individual models in the ensemble.

As demonstrated in Figure 7, the logits exhibit similar scales and distributions. While the figure presents only the CIFAR100 dataset and a single class, the figures for other datasets and classes exhibit similar structures. Based on this observation, we conclude that individual models contribute equally to the final OOD score, thereby improving the robustness of the OOD detection score estimation.

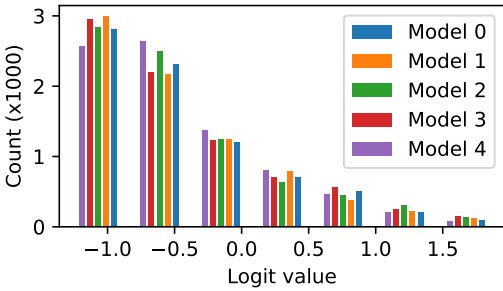

Figure 7: Logit distribution for individual models in the SDDE ensemble for the first class in CIFAR100.

As our baselines use probability averaging aggregation, we ablate logit aggregation for SDDE. In this experiment, we apply the proposed MAL aggregation to all baseline methods. The results are presented in Table 10. Our findings show that MAL improves the quality of the baseline models. At the same time, SDDE achieves higher scores in 6 out of 8 comparisons. The comparison of methods trained with OOD data is presented in Table 11. It can be seen that SDDE$_{OOD}$ achieves the highest scores in 5 out of 6 comparisons.

Table 10: Near / Far OOD detection AUROC comparison for all methods with MAL aggregation.

| Dataset | DE | NCL | ADP | DICE | SDDE (our) |
|---|---|---|---|---|---|
| MNIST | 93.45 / 99.33 | 93.69 / 99.35 | 93.90 / 99.28 | 93.88 / 99.24 | **96.24 / 99.70** |
| CIFAR10 | 91.66 / 94.36 | 91.67 / 94.24 | 91.08 / 94.02 | 90.44 / 93.64 | **91.90 / 94.55** |
| CIFAR100 | 83.36 / 82.03 | 83.63 / 81.89 | 83.04 / 81.55 | 83.62 / **83.40** | **83.69** / 82.39 |
| ImageNet | 75.21 / 85.80 | 74.89 / 85.88 | **75.82** / 84.03 | N/A | 75.26 / **85.98** |

Table 11: Comparison of OOD score aggregation methods for the SDDE$_{OOD}$ method.

| Method | CIFAR10 | | CIFAR100 | | ImageNet200 | |
|---|---|---|---|---|---|---|
| | Near | Far | Near | Far | Near | Far |
| OE-Ensemble$_{MSP}$ | 96.25 ± 0.14 | 97.34 ± 0.14 | 89.40 ± 0.02 | **88.34 ± 0.72** | 84.05 ± 0.19 | 88.77 ± 0.37 |
| OE-Ensemble$_{MAL}$ | **96.27 ± 0.01** | 97.40 ± 0.19 | 89.59 ± 0.04 | 84.41 ± 0.97 | 84.41 ± 0.15 | 89.45 ± 0.26 |
| **SDDE$_{MSP}$** | **96.27 ± 0.07** | 95.57 ± 0.05 | 89.64 ± 0.17 | 85.38 ± 1.76 | 84.10 ± 0.10 | 88.95 ± 0.15 |
| **SDDE$_{MAL}$** | 96.22 ± 0.08 | **97.60 ± 0.08** | **89.70 ± 0.22** | 85.47 ± 1.76 | **84.46 ± 0.07** | **89.57 ± 0.11** |

# F    SALIENCY MAPS VS FEATURE DIVERSITY

In Section 3, we established a link between the diversity observed in saliency maps space and the diversity in the prediction space of models. However, an unresolved question remains: does the diversity in saliency maps contribute to diversity in the feature space? To address this, we analyzed

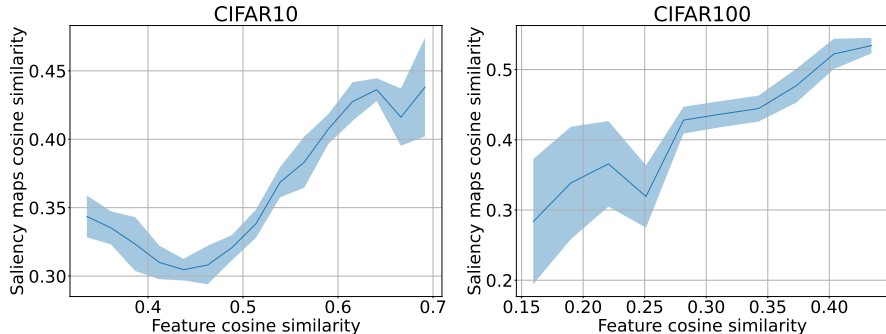

Figure 8: The dependency of ensemble saliency maps cosine similarity on feature cosine similarities. Saliency maps are computed using GradCAM. Mean and STD values w.r.t. multiple training seeds for the CIFAR10/100 datasets are reported.

the cosine similarities between the saliency maps and the models' penultimate layers' features. The findings are depicted in Figure 8. It's evident from the data that higher similarities in the feature space correspond to increased similarities in saliency maps. Consequently, we infer that enhancing the diversity of saliency maps compels ensemble models to utilize distinct features for making predictions.

## G  COMPUTATIONAL COMPLEXITY

We have conducted a comparative analysis of the computational cost of SDDE training compared to other methods. According to our measurements on the NVIDIA V100 GPU, with training details described in Section 5.1 on CIFAR10, training a single SDDE model takes 2.9 hours using a single V100 GPU. Deep Ensemble takes 6.4 hours, and Deep Ensemble without adversarial loss takes 1.4 hours. The inference time is not affected by diversity loss and remains identical for all ensemble methods. Similarly, during training, SDDE, Deep Ensemble, and Deep Ensemble without adversarial loss consume 6.2 GB, 6.8 GB, and 3.9 GB of GPU memory, respectively.

## H  SDDE TRAINING ALGORITHM

SDDE training algorithm is presented in Listing 1. During cross-entropy (CE) loss computation, each model receives its own batch. The diversity loss is computed for a single batch, forwarded through each model. The training procedure with OOD sample is presented in Listing 2. The loss is extended with Outlier Exposure (OE) objective Hendrycks et al. (2018), computed on the OOD batch. Similar to OE training, both in-distribution and OOD batches are concatenated and processed via single forward pass. This way, batch normalization statistics are adapted to OOD data.

---

**Algorithm 1** SDDE training

---

**Input:** $\mathcal{D}, N, T, \lambda, \epsilon$
**Output:** Trained weights $\theta_k, k \in \overline{1, N}$

1: Initialize $\theta_k, k \in \overline{1, N}$
2: **for** $i \leftarrow 1$ to $T$ **do**
3:     $l \leftarrow 0$
4:     **for** $k \leftarrow 1$ to $N$ **do**
5:         Sample batch $(x_k^i, y_k^i)$ from $\mathcal{D}$
6:         $l \leftarrow l + \frac{1}{N}\mathcal{L}_{CE}(x_k^i, y_k^i; \theta_k)$
7:     **end for**
8:     Sample batch $(\hat{x}_k, \hat{y}_k)$ from $\mathcal{D}$
9:     $l \leftarrow l + \lambda\mathcal{L}_{div}(\hat{x}_k, \hat{y}_k; \theta_1, \ldots, \theta_N)$
10:     **for** $k \leftarrow 1$ to $N$ **do**
11:         $\theta_k \leftarrow \theta_k - \epsilon\nabla_{\theta_k} l$
12:     **end for**
13: **end for**

---

**Algorithm 2** SDDE training with OOD data

---

**Input:** $\mathcal{D}, \mathcal{D}_{OOD}, N, T, \lambda, \epsilon$
**Output:** Trained weights $\theta_k, k \in \overline{1, N}$

1: Initialize $\theta_k, k \in \overline{1, N}$
2: **for** $i \leftarrow 1$ to $T$ **do**
3:     $l \leftarrow 0$
4:     **for** $k \leftarrow 1$ to $N$ **do**
5:         Sample batch $(x_k^i, y_k^i)$ from $\mathcal{D}$
6:         Sample unlabeled batch $(\overline{x}_k^i)$ from $\mathcal{D}_{OOD}$
7:         $l \leftarrow l + \frac{1}{N}\mathcal{L}_{CE}(x_k^i, y_k^i; \theta_k) + \frac{\beta}{N}\mathcal{L}_{OE}(\overline{x}_k^i; \theta_k)$
8:     **end for**
9:     Sample batch $(\hat{x}_k, \hat{y}_k)$ from $\mathcal{D}$
10:     $l \leftarrow l + \lambda\mathcal{L}_{div}(\hat{x}_k, \hat{y}_k; \theta_1, \ldots, \theta_N)$
11:     **for** $k \leftarrow 1$ to $N$ **do**
12:         $\theta_k \leftarrow \theta_k - \epsilon\nabla_{\theta_k} l$
13:     **end for**
14: **end for**

---

