# OpenReview forum: "Diversifying Deep Ensembles: A Saliency Map Approach for Enhanced OOD Detection, Calibration, and Accuracy"
_ICLR.cc/2024/Conference — Submitted to ICLR 2024_

### Official Review · Reviewer_T1LE · 2023-10-29

**Soundness:** 3 good
**Presentation:** 2 fair
**Contribution:** 1 poor
**Rating:** 3
**Confidence:** 3

**Summary:**

This paper introduces Saliency-Diversified Deep Ensembles (SDDE) as a new diversification technique. This method uses saliency maps produced by GradCAM and makes them as different as possible. Specifically, it computes the cosine similarities between the saliency maps and uses their mean as the diversity loss function. SDDE performs better than previous ensemble methods. In addition, it performs well in OOD detection.

**Strengths:**

- The proposed method is very simple and easy to understand.
- It shows good performance on the OpenOOD benchmark.

**Weaknesses:**

- W1. SDDE performs only with CNNs because it GradCAM that is applied to CNN layers.
- W2. The main hypothesis that ensemble diversity is proportionally related to the diversity of saliency maps, might be invalid for other datasets. The validation process proposed in the paper might not work for other cases.
- W3. SDDE can be applied to classification algorithms because CAMs are computed on the predicted classes.
- W4. Table 1 is misleading. The authors adopted additional diversity metrics, but it is obvious that their method looks to have better scores because they specifically added an additional loss function for diversity, which is based on cosine similarity.
- W5. The paper requires re-writing. The final method named SDDE_{OOD} is presented at the end of the paper, right before Section 6. The authors should describe this final method in Section 4.
- W6. It looks like MAL (Maximum Average Logit) is one of the main contributions of this paper. However, there is not enough analysis on this.

**Questions:**

What is the total training time of the entire framework when compared to previous approaches? I think it takes more time and has more FLOPs because of the CAMs computation.

---

> ### Author Response · Authors · 2023-11-14
>
> We express our sincere gratitude for the time you have taken to thoroughly review our manuscript. We likewise value the insightful critiques you have given, which we believe will sharpen the quality of our research. As we respond, we address each of your points. Should any further concerns persist or emerge, we are open for an ongoing discourse.
>
> > **[Q1]**. *SDDE performs only with CNNs because it GradCAM that is applied to CNN layers.*
> >
>
> **[A1]**. We appreciate your comment regarding the application of SDDE solely with CNNs due to the use of GradCAM which is typically applied to CNN layers. However, we would like to highlight that there are saliency maps for transformers [1] and GradCAM, being a versatile method for visualizing feature maps, can be applied to features of any dimensionality.
>
> In the context of computer vision tasks, which is our focus in this study, CNNs continue to be a stalwart choice [2] due to their strong performance, popularity in the research community, and computational efficiency. They are often favored for their speed compared to alternatives, making them a viable and relevant model to use in our studies with SDDE and GradCAM. Moreover, CNN architectures are a part of OpenOOD benchmark, which we follow in our paper. We hope this clarifies our choice of models.
>
> > **[Q2]**. *The main hypothesis that ensemble diversity is proportionally related to the diversity of saliency maps, might be invalid for other datasets. The validation process proposed in the paper might not work for other cases.*
> >
>
> **[A2]**. Thank you for your insights. We acknowledge the concern regarding the validity of our hypothesis across varied datasets. However, our work is based on the well-tested OpenOOD benchmark, ensuring diverse domain coverage.
>
> > **[Q3]**. *SDDE can be applied to classification algorithms because CAMs are computed on the predicted classes.*
> >
>
> **[A3]**. We agree with the remark about the applicability of SDDE to classification algorithms due to the computation of CAMs on predicted classes. Moreover, in our work we follow existing open-source benchmark for OOD detection (OpenOOD [4]), that is constituted of image classification tasks. Furthermore, we'd like to note that there exist GradCAM-based saliency maps methods adaptable for various tasks e.g. regression, pose estimation [3]. This proves that our approach is not limited by classification-based tasks.
>
> > **[Q4]**. *Table 1 is misleading. The authors adopted additional diversity metrics, but it is obvious that their method looks to have better scores because they specifically added an additional loss function for diversity, which is based on cosine similarity.*
> >
>
> **[A4]**. Thank you for your valuable feedback. We want to clarify that all of the baselines we considered (NCL, APD, DICE) also introduce some form of loss function to diversify their predictions, and we have not modified these baselines with our loss function.
>
> The main purpose of Table 1 is to provide a comparison among the diversity metrics of different methods. Our method aims to diversify saliency maps, which is different from most existing methods that increase the diversity in the prediction space. While diversification in the prediction space often has a trade-off with performance, our approach has shown that diversifying features can achieve balanced performance. We hope this clarifies our position and sheds light on the rationale underlying our approach.

---

> > ### Author Response · Authors · 2023-11-14
> >
> > > **[Q5]**. *The paper requires re-writing. The final method named SDDE_{OOD} is presented at the end of the paper, right before Section 6. The authors should describe this final method in Section 4.*
> > >
> >
> > **[A5]**. Thank you for your opinion. We would like to emphasize that SDDE_{OOD} is just a one piece of our pack of experiments. Consequently, we want to get attention of a reader to saliency map diversification without external data as the main topic of out work.
> >
> > > **[Q6]**. *It looks like MAL (Maximum Average Logit) is one of the main contributions of this paper. However, there is not enough analysis on this.*
> > >
> >
> > **[A6]**. We appreciate your comments and would like to clarify that our paper does not focus on  proposing MAL. We simply adopt it for ensembles and provide an ablation study on its advantages in Appendix E.
> >
> > > **[Q7]**. *What is the total training time of the entire framework when compared to previous approaches? I think it takes more time and has more FLOPs because of the CAMs computation.*
> > >
> >
> > **[A7]**. You rightly pointed out that our approach raises questions about its computational cost compared to other methods for improving ensemble diversity. To address this concern, we have conducted a comparative analysis of the computational cost of SDDE training compared to other methods. According to our measurements on the NVIDIA V100 GPU, with training details described in Section 5.1, training a single SDDE model takes 2.9 hours using a single V100 GPU. Deep Ensemble takes 6.4 hours, and Deep Ensemble without adversarial loss takes 1.4 hours. The inference time is not affected by diversity loss and remains identical for all ensemble methods. Similarly, during training, SDDE, Deep Ensemble, and Deep Ensemble without adversarial loss consume 6.2 GB, 6.8 GB, and 3.9 GB of GPU memory, respectively. We have included this information to our manuscript (see Appendix G).
> >
> > We kindly ask you to reconsider our paper, which in our view makes significant strides in the field of ensemble diversification and out-of-distribution detection. We're excited to hear any additional thoughts you might have on our revised text.
> >
> > **Reference**
> >
> > [1] Byun, Seok-Yong, and Wonju Lee. "ViT-ReciproCAM: Gradient and Attention-Free Visual Explanations for Vision Transformer." *arXiv preprint arXiv:2310.02588* (2023).
> >
> > [2] Woo, Sanghyun, et al. "Convnext v2: Co-designing and scaling convnets with masked autoencoders." *Proceedings of the IEEE/CVF Conference on Computer Vision and Pattern Recognition*. 2023.
> >
> > [3] Shavit, Yoli, Ron Ferens, and Yosi Keller. "Paying attention to activation maps in camera pose regression." *arXiv preprint arXiv:2103.11477* (2021).
> >
> > [4] Zhang, Jingyang, et al. "OpenOOD v1. 5: Enhanced Benchmark for Out-of-Distribution Detection." *arXiv preprint arXiv:2306.09301* (2023).

---

### Official Review · Reviewer_iYeu · 2023-10-30

**Soundness:** 3 good
**Presentation:** 3 good
**Contribution:** 3 good
**Rating:** 5
**Confidence:** 3

**Summary:**

In this paper, the authors propose a Saliency-Diversified Deep Ensembles (SDDE) method for classification and OOD detection. Different from previous works which often focus on diversifying the model output, the proposed method aims to diversify the feature space for improving model performance. Specifically, SDDE leverages distinct input features for predictions via computing saliency maps and applying a loss function for diversification.

**Strengths:**

The idea of using saliency to enhance the diversity of input features for OOD detection is interesting.

**Weaknesses:**

1.	The authors followed the experimental setup and training procedure from the OpenOOD benchmark (Zhang et al., 2023). I am confused as to why they did not also follow the same evaluation setup from the OpenOOD.
2.	The authors miss several state-of-the-art OOD methods [1-4] for comparison.
[1] Yue Song, Nicu Sebe, and Wei Wang. Rankfeat: Rank-1 feature removal for out-of-distribution detection. NIPS 2022.
[2] Andrija Djurisic, Nebojsa Bozanic, Arjun Ashok, and Rosanne Liu. Extremely simple activation shaping for out-of-distribution detection. ICLR 2023.
[3] Jinsong Zhang, Qiang Fu, Xu Chen, Lun Du, Zelin Li, Gang Wang, xiaoguang Liu, Shi Han, and Dongmei Zhang. Out-of-distribution detection based on in-distribution data patterns memorization with modern hopfield energy. ICLR 2023.
[4] Yiyou Sun, Yifei Ming, Xiaojin Zhu, and Yixuan Li. Out-of-distribution detection with deep nearest neighbors. ICML, 2022.
3. In Table 2, why does the proposed method show inferiority on the MINIST dataset while achieving superior performance on the rest of the datasets?

**Questions:**

Please see weaknesses.

---

> ### Author Response · Authors · 2023-11-14
>
> We sincerely appreciate the time and effort you've dedicated to reviewing our paper. As authors, we are equally grateful for the insightful feedback you provided concerning our work. In our subsequent response, we will address all the concerns. Our primary objective is to improve the quality and impact of our research. If any issues persist or new ones arise, we are ready to engage in ongoing dialogue to ensure the manuscript's refinement.
>
> > **[Q1]**. *The authors followed the experimental setup and training procedure from the OpenOOD benchmark (Zhang et al., 2023). I am confused as to why they did not also follow the same evaluation setup from the OpenOOD.*
> >
>
> **[A1]**. We would like to kindly inform you that we have strictly followed the evaluation procedure delineated in OpenOOD v1.5 [1] for conducting our research. The only modification we have implemented is the multi-seed evaluation for ImageNet. We believe this slight adjustment is necessary for the depth and accuracy of our study.
>
> > **[Q2]**. *The authors miss several state-of-the-art OOD methods [1-4] for comparison. [1] Yue Song, Nicu Sebe, and Wei Wang. Rankfeat: Rank-1 feature removal for out-of-distribution detection. NIPS 2022. [2] Andrija Djurisic, Nebojsa Bozanic, Arjun Ashok, and Rosanne Liu. Extremely simple activation shaping for out-of-distribution detection. ICLR 2023. [3] Jinsong Zhang, Qiang Fu, Xu Chen, Lun Du, Zelin Li, Gang Wang, xiaoguang Liu, Shi Han, and Dongmei Zhang. Out-of-distribution detection based on in-distribution data patterns memorization with modern hopfield energy. ICLR 2023. [4] Yiyou Sun, Yifei Ming, Xiaojin Zhu, and Yixuan Li. Out-of-distribution detection with deep nearest neighbors. ICML, 2022.*
> >
>
> **[A2]**. In our paper, we have proposed a novel ensembling method and focused on comparisons with other ensemble approaches. The mentioned methods, to our understanding, target single-model designs, and have already been implemented in the OpenOOD 1.5 paper. As we run the same evaluation pipeline as in the OpenOOD benchmark, our results can be compared with the original OpenOOD paper. Thus, our approach outperforms the ASH [2] method, which is included in OpenOOD (C10 Near: 75.27% → 92.06%, C10 Far: 78.49% → 94.60%, C100 Near: 78.20% → 92.06%, C100 Far: 80.58% → 94.60%). We hope this elucidates our considerations regarding the methods we've chosen to compare in our study.
>
> We hope this explanation addresses the queries raised and provides sufficient insight into our methodological choices. Given the clarified points stated above, we cordially request that you reconsider the evaluation of our paper.
>
> **Reference**
>
> [1] Zhang, Jingyang, et al. "OpenOOD v1. 5: Enhanced Benchmark for Out-of-Distribution Detection." arXiv preprint arXiv:2306.09301 (2023).
>
> [2] Djurisic, Andrija, et al. "Extremely simple activation shaping for out-of-distribution detection." *arXiv preprint arXiv:2209.09858*(2022).

---

### Official Review · Reviewer_FdHh · 2023-11-01

**Soundness:** 2 fair
**Presentation:** 2 fair
**Contribution:** 2 fair
**Rating:** 3
**Confidence:** 4

**Summary:**

This paper suggests that deep ensembles are less effective due to the homogeneity of learned patterns. So, the authors try to diversify the saliency maps of the models involved.

By doing so, the paper claims to attain SOTA results.

**Strengths:**

+ Good results
+ Clearly written

**Weaknesses:**

- As per my understanding, saliency maps should highlight the object regions to help classification. If we make them highlight different regions, as done in Fig.1, it defeats the purpose of saliency maps. I don't agree with the idea that we should diversify saliency maps spatially, to the extent they start highlighting backgrounds.
-Technical contributions are very limited.

**Questions:**

Why do authors think diversifying saliency maps is the same as diversifying features?

---

> ### Author Response · Authors · 2023-11-14
>
> Thank you for reviewing our manuscript and acknowledging its significance. We appreciate your valuable feedback and will address your concerns in detail. Our aim is to improve our work's quality and we are open to further discussion to resolve any remaining issues.
>
> > **[Q1].** *As per my understanding, saliency maps should highlight the object regions to help classification. If we make them highlight different regions, as done in Fig.1, it defeats the purpose of saliency maps. I don't agree with the idea that we should diversify saliency maps spatially, to the extent they start highlighting backgrounds. -Technical contributions are very limited.*
> >
>
> **[A1].** We respect your perspective on the traditional function of saliency maps; however, we would like to offer a different viewpoint supported by our experimental results. Our approach, termed SDDE, might appear unconventional at first glance, but it's crucial to note that it doesn't compromise the performance of individual models. This is clearly demonstrated in Table 1 of our manuscript, where the accuracy of individual models remains intact despite the diversification of saliency maps. We thus conclude that even models, focusing on backgrounds and separate parts of the image, can achieve competitive prediction accuracy.
>
> > **[Q2].** *Why do authors think diversifying saliency maps is the same as diversifying features?*
> >
>
> **[A2].** Thank you for your thoughtful question. To address your query, we conducted an analysis of the relationship between the diversity in saliency maps and the diversity in the features of the models' penultimate layers. This involved measuring the cosine similarities between the saliency maps and these features. The results of this analysis are presented in Figure 8 of Appendix F of our manuscript.
>
> Our findings indicate a clear correlation: as similarities in the feature space increase, so do the similarities in saliency maps. This observation led us to conclude that by diversifying the saliency maps, we indirectly encourage the ensemble models to utilize a broader range of features for their predictions. This inference bridges the gap between the diversification of saliency maps and features in our approach. We understand the importance of this clarification and have thus incorporated the relevant details into our manuscript (Appendix F).
>
> Thank you once again for your valuable feedback, which has helped us improve the clarity and depth of our research. We kindly ask you to reconsider your evaluation of our manuscript, taking into account the additional information provided regarding our methods and results. We look forward to your continued input and further discussions.

---

### Official Review · Reviewer_WdEW · 2023-11-02

**Soundness:** 2 fair
**Presentation:** 3 good
**Contribution:** 1 poor
**Rating:** 3
**Confidence:** 4

**Summary:**

This paper presents SDDE, an ensembling method for classification and OOD detection. SDDE forces the models within the ensemble to use different input features for prediction, which increases ensemble diversity. Improved confidence estimation and OOD detection make SDDE a useful tool for risk-controlled recognition. SDDE is further generalised for training with OOD data and achieved SOTA results on the
OpenOOD benchmark.

**Strengths:**

Originality: The new aspect in the paper is actually that the diversity loss is combined with cross-entropy during training in a single optimization objective.

Quality: The paper structure seems adequate. The balance between the theory and experiments seems adequate. The proposed method has been examined and compared against the other state of the art technologies. This paper presents rich ablation results.

Clarity: The proposed method sounds reasonable and easy to follow.

Significance: The paper shows that a large number of experiments have been achieved with comprehensive discussion.

**Weaknesses:**

Originality: The diversity loss is combined with cross-entropy during training in a single optimization objective. This additional component sounds like an incremental change. More deep investigation on the incentive of using this combination is required.

Quality: The discussion on the weaknesses of the proposed method seems missing.

Clarity: This paper does not present sufficient explanation to the introduction of the combination of diversity loss and cross-entropy. The introduced strategy sounds like adhoc solution and requires wide discussion on the underlying mechanism.

Significance: The proposed method does not significantly outperforms the other state of the art technologies. In some of the metrics, the proposed method seems to work well but not all or large metrics.

**Questions:**

1. Why the combination of diversity loss and cross-entropy is the best way to take on board?
2. To explain the convergence property of the combined solution in the paper.
3. To provide computational complexity analysis of the compared algorithms.

**Details Of Ethics Concerns:**

no.

---

> ### Author Response · Authors · 2023-11-14
>
> Thank you for your insightful feedback regarding our manuscript. We appreciate your comments on our work, and we would like to address your concerns and clarify all the aspects of our approach.
>
> > **[Q1].** *Quality: The discussion on the weaknesses of the proposed method seems missing.*
> >
>
> **[A1].** Thank you for pointing out the importance of discussing limitations of the proposed method. We follow this guideline and cover possible drawbacks of our approach in Section 7 of the submitted manuscript. Furthermore, in order to make this part of our work more distinguishable, we have made a separate section on the limitations (see Section 7 of the updated manuscript).
>
> > **[Q2].** *Significance: The proposed method does not significantly outperforms the other state of the art technologies. In some of the metrics, the proposed method seems to work well but not all or large metrics.*
> >
>
> **[A2].** Thank you for your feedback. While we respect your view, we politely disagree with the assessment. In our paper, we have proposed a novel ensembling method and focused on comparisons with other ensemble approaches. Our results indicate meaningful improvements over current methods, specifically in the area of OOD detection tasks, even on large-scale datasets; please, refer to Tables 3 and 4 Near/Far OOD detection scores: C10 Near: 91.22% → 92.06%, C10 Far: 93.55% → 94.60%, C100 Near: 83.18% → 83.65%, C100 Far: 82.15% → 82.39%, ImageNet Far: 82.85% → 85.98%. We believe the margin of improvement we demonstrate is significant and worthy of consideration. It's our hope that this will be taken into account in the evaluation.
>
> > **[Q3].** *To explain the convergence property of the combined solution in the paper.*
> >
>
> **[A3].** Our research leverages a well-established setup for optimizers and models using SGD and ResNets, which is grounded in a significant body of previous research. With respect to your question on convergence properties, this is still an open problem in the research community and it transcends the scope of our study [4]. While it is a critically important topic, many pragmatic studies, including ours, do not go in-depth into this subject because the focus is primarily on real-world applications and empirical approach. We appreciate your understanding of these constraints and your important role in helping us improve our contributions in this field.

---

> > ### Author Response · Authors · 2023-11-14
> >
> > > **[Q4].** *To provide computational complexity analysis of the compared algorithms.*
> > >
> >
> > **[A4].** You rightly pointed out that our approach raises questions about its computational cost compared to other methods for improving ensemble diversity. To address this concern, we have conducted a comparative analysis of the computational cost of SDDE training compared to Deep Ensembles. According to our measurements on the NVIDIA V100 GPU, with training details described in Section 5.1, training a single SDDE model takes 2.9 hours using a single V100 GPU. Deep Ensemble takes 6.4 hours, and Deep Ensemble without adversarial loss takes 1.4 hours. The inference time is not affected by diversity loss and remains identical for all ensemble methods. Similarly, during training, SDDE, Deep Ensemble, and Deep Ensemble without adversarial loss consume 6.2 GB, 6.8 GB, and 3.9 GB of GPU memory, respectively. We have included this information to our manuscript (see Appendix G).
> >
> > > **[Q5].** *Originality: The diversity loss is combined with cross-entropy during training in a single optimization objective. This additional component sounds like an incremental change. More deep investigation on the incentive of using this combination is required.* **[Q5*].** *Why the combination of diversity loss and cross-entropy is the best way to take on board?*
> > >
> >
> > **[A5].** It is indeed a common practice to utilize cross-entropy in classification tasks. However, the introduction of the SDDE loss is not a mere incremental improvement to CE. Our approach fundamentally alters the optimization landscape by integrating a diversification component, which is not typically considered in standard CE-based methods. This integration goes beyond the conventional usage of CE and contributes to the novelty of our work. The concept of introducing diversification losses is indeed known in the context of ensemble methods [1, 2, 3]. However, our work presents a novel perspective on this concept. We have developed a unique approach to diversification that differs from traditional methods used in ensemble models.
> >
> > Considering your valuable feedback and the clarifications we've provided about our approach and its impact, we kindly request that you reassess the evaluation of our manuscript.
> >
> > [1] Changjian Shui, Azadeh Sadat Mozafari, Jonathan Marek, Ihsen Hedhli, and Christian Gagn ́e. Di-versity regularization in deep ensembles. 2018.
> >
> > [2] Tianyu Pang, Kun Xu, Chao Du, Ning Chen, and Jun Zhu. Improving adversarial robustness via promoting ensemble diversity. In International Conference on Machine Learning, pp. 4970–4979. PMLR, 2019.
> >
> > [3] Alexandre Ram ́e and Matthieu Cord. Dice: Diversity in deep ensembles via conditional redundancy adversarial estimation. In ICLR 2021-9th International Conference on Learning Representations, 2021.
> >
> > [4] Abdulkadirov, R.; Lyakhov, P.; Nagornov, N. Survey of Optimization Algorithms in Modern Neural Networks. Mathematics 2023, 11, 2466. https://doi.org/10.3390/math11112466
> >
> > [5] Zhang, Jingyang, et al. "OpenOOD v1. 5: Enhanced Benchmark for Out-of-Distribution Detection." *arXiv preprint arXiv:2306.09301* (2023).

---

### Meta-Review · Area_Chair_u9b9 · 2023-12-05

**Metareview:**

This papers aims to improve deep ensembles through the addition of a regularizer that encourages diversity amongst the component models’ GradCAM saliency maps. Although the authors addressed concerns in the rebuttal phase, there are still several outstanding concerns. First there appears to be a lack of comparison to other OOD detection methods, and the authors limit their evaluation to CNNs and a single benchmark. There were also concerns about the computational costs of the proposed method and the justification for enforcing diversity on saliencey maps; however, the authors address these concerns in their revision. Nevertheless, this paper could benefit from additional experiments (more architectures/tasks/datasets) before being considered for publication at a top-tier conference.

**Justification For Why Not Higher Score:**

The reviewers did not engage with the authors during the rebuttal period, which is a shame. Nevertheless, the concerns mentioned by the reviewers were - in my opinion - not sufficiently addressed by the authors.

**Justification For Why Not Lower Score:**

N/A

---

### Decision · Program_Chairs · 2024-01-16

Reject